# Influenza Vaccination in Italian Healthcare Workers (2018–2019 Season): Strengths and Weaknesses. Results of a Cohort Study in Two Large Italian Hospitals

**DOI:** 10.3390/vaccines8010119

**Published:** 2020-03-05

**Authors:** Donatella Panatto, Piero Luigi Lai, Stefano Mosca, Elvina Lecini, Andrea Orsi, Alessio Signori, Silvana Castaldi, Elena Pariani, Laura Pellegrinelli, Cristina Galli, Giovanni Anselmi, Giancarlo Icardi

**Affiliations:** 1Department of Health Sciences, University of Genoa, Via Pastore 1, 16132 Genoa, Italy; pierolai@unige.it (P.L.L.); elvinalecini@gmail.com (E.L.); andrea.orsi@unige.it (A.O.); alessio.signori.unige@gmail.com (A.S.); icardi@unige.it (G.I.);; 2Interuniversity Research Center on Influenza and Other Transmissible Infections (CIRI-IT), Via Pastore 1, 16132 Genoa, Italy; stefanom@unige.it (S.M.); Elena.Pariani@unimi.it (E.P.); 3Department of Biomedical Sciences for Health, University of Milan, Via C. Pascal 36, 20133 Milano, Italy; silvana.castaldi@unimi.it (S.C.); laura.pellegrinelli@unimi.it (L.P.); cristina.galli@unimi.it (C.G.); giovanni.anselmi@unimi.it (G.A.); 4Fondazione IRCCS Ca’ Granda Ospedale Maggiore Policlinico di Milano, Via F. Sforza 28, 20122 Milano, Italy; 5Ospedale Policlinico San Martino IRCCS, Largo R. Benzi 10, 16132 Genoa, Italy

**Keywords:** influenza, influenza vaccination, healthcare workers, laboratory-confirmed influenza, influenza vaccination coverage

## Abstract

Background: Annual vaccination is the most effective way to combat influenza. As influenza viruses evolve, seasonal vaccines are updated annually. Within the European project Development of Robust and Innovative Vaccine Effectiveness (DRIVE), a cohort study involving Italian healthcare workers (HCWs) was carried out during the 2018-2019 season. Two aims were defined: to measure influenza vaccine effectiveness (IVE) against laboratory-confirmed influenza cases and to conduct an awareness-raising campaign to increase vaccination coverage. Methods: Each subject enrolled was followed up from enrollment to the end of the study. Each HCW who developed ILI was swabbed for laboratory confirmation of influenza. Influenza viruses were identified by molecular assays. A Cox regression analysis, crude and adjusted for confounding variables, was performed to estimate the IVE. Results: Among the 4483 HCWs enrolled, vaccination coverage was 32.5%, and 308 ILI cases were collected: 23.4% were positive for influenza (54.2% A(H1N1) pdm09; 45.8% A(H3N2)). No influenza B viruses were detected. No overall IVE was observed. Analyzing the subtypes of influenza A viruses, the IVE was estimated as 45% (95% CI: -59 to 81) for A(H1N1) pdm09. Conclusions: Vaccination coverage among HCWs increased. Study difficulties and the circulation of drifted variants of A(H3N2) could partly explain the observed IVE.

## 1. Introduction

Influenza is an acute respiratory illness of global importance. Every year, 350 million to 1 billion cases, approximately 3 to 5 million cases of severe illness, and between 290,000 and 650,000 deaths are estimated worldwide [1]. In Europe, a recent study showed that influenza was the infectious disease with the greatest impact on the population’s health, with a burden of 81.8 disability-adjusted life years (DALYs) per 100,000 persons (95% Uncertainty Interval (UI): 76.9–86.5) [2]. The European Centre for Disease Prevention and Control (ECDC) estimates that influenza causes about 40,000 premature deaths each year in the European Union [3].

In Italy, during the 2018–2019 influenza season, about 8 million of Influenza-Like Illness (ILI) cases were registered, 31.7% of which were caused by influenza viruses [4,5].

Although anyone can catch influenza, some groups are at greater risk than others. Categories at greater risk of severe disease or complications are pregnant women, children under the age of 5 years, the elderly, individuals with chronic medical conditions (such as chronic cardiac, pulmonary, renal, or metabolic diseases) and immunosuppressed individuals (such as those with HIV/AIDS, cancer patients, and those on chemotherapy or steroids) [1,6].

Healthcare workers (HCWs) have a high risk of catching influenza, owing to their frequent exposure to infected patients, and risk spreading the infection further, particularly to vulnerable individuals [6,7,8,9].

Annual vaccination is the most effective way to prevent influenza and to reduce its clinical and economic burden. Indeed, the World Health Organization (WHO) and other international health authorities recommend vaccination for high-risk groups, including HCWs [1,6,10]. Accordingly, Italy has a national program of free influenza vaccination, which targets high-risk categories, including HCWs [11,12]; its optimal target is to achieve 95% vaccination coverage, while 75% coverage is its minimum goal. Despite the strong recommendations provided by international and national organizations, seasonal influenza coverage rates among HCWs in Europe and Italy are usually low [7,8,9,13,14]. Increasing seasonal influenza vaccination coverage in high-risk categories is therefore a priority issue.

Because influenza viruses continuously evolve, seasonal influenza vaccines are updated annually in order to achieve a good match between vaccine strains and strains circulating in the population. In recent years, several studies have highlighted the need to carry out annual effectiveness studies in order to obtain solid data and to better understand the role of the variables that influence the effectiveness of influenza vaccines [15,16,17,18,19,20,21,22].

Several influenza vaccines were licensed in Italy in the 2018-2019 season: conventional trivalent vaccines, one adjuvanted trivalent vaccine and quadrivalent vaccines [12]. In particular, according to annual influenza recommendations in Italy, quadrivalent vaccines are indicated as preferable for HCW immunization [12].

Within the European project DRIVE (Development of Robust and Innovative Vaccine Effectiveness) 2018–2019 [15], a cohort study to measure seasonal influenza vaccine effectiveness (IVE) against laboratory-confirmed influenza cases in Italy was carried out during the 2018–2019 season. This study examined a cohort of HCWs (subjects ≥ 18 years) working in two large Italian hospitals (Genoa and Milan).

Another objective was to conduct a tailor-made pro-vaccination awareness-raising campaign aimed at HCWs in order to increase their unsatisfactory vaccination coverage.

## 2. Materials and Methods

### 2.1. Study Population and Study Design

The study protocol was approved by the Ethics Committee of the Liguria Region (Genoa, Italy) (n° 314/2018) and the Ethics Committee of Milan Area 2 (n° 1941/09.10.2018).

A new Information Technology (IT) system was developed in order to manage the data and monitor all the phases of the study. The IT system was based on the Mac OS X platform and on the Filemaker Server database. The database was accessible to CIRI-IT staff via client and web connection through private credentials (unique to each member of the CIRI-IT team), in order to guarantee the high quality and safety of the data and to insert and update any information of the study.

Two large Italian hospitals (Genoa and Milan) were involved: Hospital Policlinico San Martino Genoa and Fondazione IRCCS Ca’ Granda Ospedale Maggiore Policlinico Milan.

Hospital Policlinico San Martino Genoa is located in metropolitan area of Genoa, a city of 650,000 inhabitants. It is a tertiary teaching hospital with 1200 beds. The hospital includes more than 70 wards. The hospital is the acute care regional reference center and accounts for 55% of all hospital admissions in the Genoa metropolitan area, and 3200 HCWs work in this hospital. Hospital Policlinico San Martino is the chief seat of a school of medicine and several postgraduate schools of medicine and surgery, and it provides of more than 3200 courses of lifelong medical education per year.

Fondazione IRCCS Ca’ Granda Ospedale Maggiore Policlinico (Fondazione) is a research and teaching hospital placed in the center of Milan (a city of about 1,350,000 inhabitants). It has 912 accredited beds and 95 beds for day surgery and day hospital admissions. It is the first hospital in Italy for number of deliveries, more the 5500 with maternal and child tertiary care, and it has 16 research laboratories and 57 teaching areas. Its emergency department is devoted to general, pediatrics and obstetric care, and 2500 HCWs work in this hospital. Fondazione is the chief seat of a school of medicine, 25 postgraduate schools in medicine and surgery, and it is provider of more than 4500 courses of lifelong medical education per year.

All active HCWs working in the health facilities mentioned above were invited to participate in the cohort study.

HCWs were eligible if they agreed to participate and provided informed consent. The only exclusion criterion was having a contraindication for influenza vaccination.

Subjects were enrolled from week 40/2018 to week 51/2018. Each subject enrolled was followed up from the time of enrollment to the end of the study (week 17/2019). Subjects left follow-up at various time-points: the occurrence of laboratory-confirmed influenza, or death, or voluntary withdrawal from the study, or at the end of the study period (week 17/2019). During the enrollment, all data (demographic characteristics, chronic conditions, risk factors, vaccination status (date of vaccination and vaccine brand)) were collected by means of a questionnaire (paper and electronic) specifically designed for this study (Appendix A).

The CIRI-IT team maintained constant contact with enrolled subjects through communication systems (telephone) and IT systems, and a reminder was sent weekly by email or text message.

Enrolled subjects who developed ILI according to the ECDC case definition [23] and contacted the CIRI-IT staff <8 days after symptom onset were deemed eligible. A respiratory swab was taken from each ILI case in accordance with a pre-defined protocol for laboratory confirmation of influenza. At the same time, clinical data were collected by means of a questionnaire (Appendix A).

As repeated influenza infections during the same season are extremely rare, only the first confirmed episode of influenza was considered in this study.

### 2.2. Laboratory Methods

In accordance with national protocols, influenza viruses in respiratory samples (oro-pharyngeal swabs) were identified by means of molecular assays, as prescribed by the WHO’s updated international guidelines for influenza virus surveillance [24]. Total RNA was extracted from each respiratory swab by using a commercial kit (Invisorb^®^ Spin Virus RNA Mini kit, Stratec Biomedical AG, Germany or QIAamp^®^ Viral RNA Mini Kit, QIAGEN GmbH, Germany) according to the manufacturer’s instructions. The material extracted was tested for the simultaneous identification of influenza A and B viruses by means of a one-step real-time multiplex retro-transcription (RT) PCR assay targeting the matrix region and the nucleoprotein region, respectively [24]. Samples positive for influenza type A were further subtyped through a one-step real-time RT-PCR test targeting the hemagglutinin (HA) gene, in order to distinguish between A(H1N1) pdm09 and A(H3N2) subtypes. Influenza type B-positive samples were characterized into B-Yamagata and B-Victoria lineages by means of a one-step real-time multiplex RT-PCR assay [24]. To check extraction performance, amplification of the human ribonuclease P gene (RNP) was carried out at the same time; this procedure utilized a specific primer/probe set and adopted the same thermal profile as that of the influenza A/B virus real time RT-PCR assay [24]. Specific positive and negative controls were tested at the same time as the samples in order to validate each real-time RT-PCR run. Samples showing a cycle threshold (Ct) value <40 were considered positive.

Samples positive for influenza viruses were further sequenced and phylogenetically analyzed. Briefly, the material extracted was retro-transcribed and amplified in order to obtain the HA gene, as previously described [25,26]. The amplicons were purified by means of a commercial kit (NucleoSpin Gel^®^ and a PCR clean-up kit, Macherey-Nagel, Germany) and sequenced by means of the Sanger method. The edited study sequences of each subtype were aligned with the respective reference strains retrieved from the Global Initiative on Sharing All Influenza Data (GISAID) EpiFlu database [27], including the respective vaccine strains for the 2018–2019 influenza season (A(H1N1): A/Michigan/45/2015, accession number (AN) EPI849370; A(H3N2): A/Singapore/INFIMH-16-0019/2016, AN EPI1151840). Multiple sequence alignments were carried out by using the ClustalW program, implemented in the BioEdit software [28]. Nucleotide sequence identity was calculated by means of the Sequence Identity Matrix tool of BioEdit software [28]. Phylogenetic analysis was conducted by means of MEGA software, version 6.0 [29]. The study sequences were submitted to the GenBank^®^ database on the NCBI website [30] under ANs MT080793-MT080807 and MT080950-MT080973.

### 2.3. Statistical Analysis

The DRIVE project included two study designs: case–control negative test study and cohort study. For cohort studies, DRIVE recommend a minimum of 4000 enrolled subjects in order to obtain a reliable IVE estimate.

We calculated that we needed to follow up a minimum of 4224 subjects. Considering conservative parameters, 20% vaccination coverage, the incidence of ILI in this population group (18–65 years of age) (about 15%), setting the significance level at 5% and the test power at 90% and utilizing a two-tailed test, it would have been sufficient to recruit 3520 non-vaccinated persons and 704 vaccinated individuals (a total sample size of 4224 subjects) in order to observe a level of vaccine effectiveness of 30% (i.e., a 15% proportion of ILI in the non-vaccinated subjects and 10.5% in those vaccinated).

The results were summarized in frequency tables. For data analysis, subjects were stratified into 6 age groups: 18–24, 25–34, 35–44, 45–54, 55–64 and ≥65 years.

Association between vaccination status and other characteristics as region and influenza vaccination in previous season (2017–2018) were tested by mean of chi-square tests.

To estimate the IVE, a Cox regression analysis with laboratory-confirmed influenza cases as the dependent variable was performed. Vaccination status (yes/no) (2018–2019 season) was used as the independent variable and was considered time-dependent. Vaccinated subjects were considered not vaccinated from the time of enrollment up to 14 days after of the date of vaccination; from that time until the end of follow-up they were considered as part of the vaccinated group.

The Cox regression model was adjusted for gender, age group, influenza vaccination in the previous season (2017–2018) (yes/no), any chronic condition (yes/no), number of hospitalizations in the previous 12 months (yes/no) and site of enrollment (Genoa and Milan). Hazard ratios (HRs) were used to estimate the IVE as (1-HR) × 100, and both unadjusted and adjusted estimates were reported. In the regression models, the interaction terms between vaccination and the demographic and clinical characteristics were also included in order to check for differences in IVE across the various subgroups of subjects. All statistical analyses were performed by means of Stata 14 (StataCorp).

## 3. Results

A total of 4483 HCWs (2270 in Genoa and 2213 in Milan) were enrolled. The characteristics of the subjects and their vaccination status are shown in Table 1.

Females accounted for 70.5% of the subjects enrolled. The overall median age was 37 years (interquartile range (IQR): 26–52). The largest age group was that of subjects aged 25–34 years (25.9%), followed by those aged 45-54 years (21.8%). As the cohort studied consisted of active HCWs, few subjects in the ≥65 year age group were included, both in Genoa and Milan (Appendix A). The age distribution differed significantly (*P* < 0.001) between Genoa and Milan, the HCWs enrolled in Milan (median age 34 years, IQR: 25–50) being younger than those enrolled in Genoa (43 years, IQR: 27–53) (Appendix A).

The percentage of subjects with at least one underlying chronic disease increased with age; the same trend was observed in the number of hospitalizations in the previous 12 months (Table 1).

Overall vaccination coverage was 32.5%. In the Genoa hospital, a higher percentage of vaccinated HCWs was registered than in Milan hospital (36.6% vs 28.4%; *P* < 0.001). HCWs in the >65 year age group had the highest coverage rate (Appendix A).

Among subjects with at least one underlying chronic disease, vaccination coverage was 37.1%, while in HCWs without chronic conditions it was 31.7% with a statistically significant difference (*p* = 0.004).

The results regarding vaccination status in both the 2018–2019 and 2017–2018 seasons showed that about 40% of subjects vaccinated in the 2018–2019 season had also been vaccinated in the previous season, while only 2% of those unvaccinated in 2018–2019 season had been vaccinated in the 2017–2018 season (Table 2). This difference was statistically significant (*p* = 0.001).

The vaccine brand was known in 99.9% of vaccinated subjects, and 97.5% of subjects received a quadrivalent vaccine (Fluarix Tetra^®^ 52.6% or Vaxigrip Tetra^®^ 44.8%). Only 1.9% received an adjuvanted trivalent vaccine (Fluad^®^) (Table 1).

Overall, 308 ILI cases were reported (6.9%) in enrolled subjects. Among 1459 vaccinated HCWs, 157 subjects reported an ILI (10.8%), and among 3024 unvaccinated subjects, 151 reported an ILI (5.1%). According to the Italian influenza surveillance (INFLUNET) [4], in Italy the cumulative incidence of ILI during the 2018–2019 season in individuals aged 15–64 was 12.9%.

From each ILI case an oropharyngeal swab was collected, and 72 (23.4%) of these were positive for influenza viruses. All positive samples were influenza virus type A: 39 (54.2%) were A(H1N1) pdm09 and 33 (45.8%) were A(H3N2). The subtypes A(H1N1) pdm09 and A(H3N2) co-circulated, although the A(H1N1) pdm09 strain was prevalent in the first half of the epidemic season and the A(H3N2) strain in the second half. No influenza B viruses were detected (Figure 1).

Appendix A reports the factors associated with the incidence of laboratory-confirmed influenza. Patients ≥ 25 years old had a significantly higher incidence of influenza than those aged 18–24 years. This age-related pattern was observed with regard to A(H1N1) pdm09, as there was a higher incidence of influenza among subjects with at least one underlying chronic disease than among those without chronic diseases. No factors were associated with the incidence of influenza due to A(H3N2).

Of the 157 vaccinated subjects with ILI, 35 (22.3%) were positive for influenza viruses (A(H1N1) pdm09: 11; A(H3N2): 24). Of the 151 unvaccinated subjects with ILI, 37 (24.5%) were positive for influenza viruses (A(H1N1) pdm09: 28; A(H3N2): 9).

All A(H1N1) pdm09 viruses identified in this study fell into genetic group 6B, subgroup 6B.1 and represented by the vaccine virus A/Michigan/45/2015. Although strains within the 6B.1 subgroup had additional amino acid substitutions, the great majority of A(H1N1) pdm09 viruses were antigenically related to the vaccine virus A/Michigan/45/2015 [31], with which shared a high nucleotide identity.

All A(H3N2) viruses identified in this study belonged to genetic group 3C and were distributed in two clades: 3C.2a and 3C.3a. The majority (60%) of strains fell into clade 3C.2a and, particularly, in subclade 3C.2a1, as the vaccine virus A/Singapore/INFIMH-16-0019/2016, with which shared a nucleotide identity of 98.4%–98.7%. Our study strains had additional amino acid changes, thus falling into the subgroup 3C.2a1b. The remaining sequences (40%) fell in the 3C.3a clades and were drifted variants characterized by amino acid substitutions Y159S and F193S. 3C.3a viruses were identified mainly in the second part of the season and showed a nucleotide identity of 96.5%–96.7% with the vaccine virus.

No overall IVE against laboratory-confirmed influenza A was detected. When the subtypes of influenza A viruses were considered, the IVE was estimated as 45% (95% CI: −59 to 81) for A(H1N1) pdm09.

No differences in IVE were observed between the two regions (Liguria and Lombardy; p for interaction region*vaccination was 0.67 for any influenza, 0.84 for A(H1N1)pdm09 and 0.55 for A(H3N2)) nor between the two main brands (p for interaction brand*vaccination was 0.94 for any influenza, 0.23 for A(H1N1)pdm09 and 0.28 for A(H3N2)). Similarly, no other characteristics were seen to significantly affect the IVE.

## 4. Discussion

This study was included in a European project (DRIVE) aimed at measuring seasonal IVE against medically attended laboratory-confirmed influenza during 2018–2019 season [15]. The DRIVE project came in response to the European Medicines Agency (EMA) guideline on influenza vaccines [32], which requires that annually observational IVE studies be conducted in the EU/EEA as part of the post-licensure commitments of vaccine manufacturers.

We chose to conduct a cohort study among HCWs, as they constitute one of the high-risk categories for whom vaccination is strongly recommended [1,6,10,11,12]. Moreover, as they are active workers, they tend to be healthy and to have few underlying chronic diseases, a condition which minimizes negative confounders. These data are confirmed by the characteristics of our cohort; indeed, most HCWs enrolled in the study did not have any chronic conditions (*n* = 3764; 84:0%) nor had they been hospitalized in the previous 12 months (*n* = 4411; 98.3%).

Overall, no IVE against laboratory-confirmed influenza cases was observed in this cohort study. However, as some critical points emerged during the study, this result should be interpreted with caution. Although cohort studies are considered suitable epidemiological studies for collecting a significant amount of data and involve a considerable percentage of the population, they are difficult to supervise, owing to the fact that the many variables involved such as the difficulty of reaching HCWs during enrolment and their difficulty in adhering to the study protocol for the entire study period nearly 6 months can modify the results of the study [33,34].

Our cohort study therefore has some limitations as follows.

There was limited time available for implementing an efficacious awareness-raising program and for enrolling HCWs, owing to the delay in obtaining the approval of the ethics committee. To overcome this obstacle, the timeframe for enrolment was prolonged in order to enable the research group to reach at least the 4000 subjects envisioned in the planning phase. Enrolling cohort participants proved more difficult than we expected for various reasons: during working hours, HCWs do not want to be disturbed; as the hospitals are made up of separate pavilions, it was not easy to move around; as HCWs work on three shifts, completing the vaccination program in a given ward means having to return to the same place three times. Moreover, the limited time available for awareness-raising programs resulted in poor adherence to the study protocol (i.e., communicating each case of ILI to the CIRI-IT).

Regarding the probable underestimation of ILI cases, it can be observed that, if in our cohort we had had the same cumulative incidence registered by the INFLUNET surveillance system [4] (12.9%) and other studies published in the 2018–2019 season [3,5] in subjects aged 15–64 years, we would have expected about 580 ILI cases instead of the 308 reported. Moreover, considering that the vaccination coverage rate recorded in our cohort (32.5%) was three times higher than that registered in the general Italian population aged 18–64 (about 10%), we would have expected to see about 180 cases of ILI among vaccinated subjects and about 400 cases among the unvaccinated, even if we hypothesized a vaccine effectiveness level of zero. Given that only 151 (5.1%) unvaccinated subjects contacted the CIRI-IT to report ILI, it is evident that the level of under-reporting by these subjects was high; this accounts for the unreliable result regarding IVE in our study. It also points to an underestimation of IVE, both overall and by subtype. Similarly, in the case of A(H1N1) pdm09, for which some, albeit minimal, effectiveness was observed, a better result could have been obtained.

The low incidence of ILI recorded in the unvaccinated group was due to low compliance with follow-up surveillance. Although weekly reminders to notify the occurrence of ILI symptoms were sent by email to all study participants, this was not enough to guarantee compliance. The percentage of subjects who reported an ILI was higher among vaccinated HCWs than in unvaccinated subjects (10.8% vs. 5.1%), suggesting that vaccinated individuals were more willing to participate in the study than unvaccinated subjects. This issue was also noted in other cohort studies [34]. This obstacle could also be due to scant awareness of the issue of influenza among these HCWs. Indeed, inadequate awareness of influenza and of the importance of influenza vaccination among HCWs is confirmed by the low vaccination coverage rates in this category, as documented by several European and Italian studies [7,8,9,13,14,35,36,37]. In this regard, the ECDC reported a coverage rate of 15.6% among HCWs in Italy in the 2016–2017 season [35]. It must be stressed, however, that healthcare workers are a key target for influenza vaccination, not only for the purpose of protecting the individual, but also with a view to reducing the spread of influenza among vulnerable/fragile patients and to ensuring the provision of healthcare services during influenza epidemics [11].

Several studies have investigated the probable causes of low vaccination coverage among HCWs [9,36]. The main obstacles identified by these studies were: the conviction that influenza is not a serious illness (many HCWs believe that they are immune, in that they claim not to have contracted an ILI in the seasons in which they were not vaccinated, or are convinced that only elderly subjects and those with chronic pathologies are at risk); concerns about the adverse effects of vaccination; skepticism regarding vaccine efficacy; the assertion of their own autonomy/responsibility in decision-making with regard to vaccination; organizational difficulties (overlapping of working timetables with those of opening hours of vaccination clinics, etc.); and, finally, a generally negative attitude [9,36].

Another limitation of the study was that the protocol of the DRIVE study did not envision gathering information on the individual occupation of each HCW. The fact that these data are missing may have led to underestimation of the effectiveness of vaccination.

IVE varies from year to year for a variety of reasons; these include mismatch between the vaccine virus strains and the circulating strains, waning immunity and possible interference by previous vaccinations.

Studies conducted by other authors have found moderate IVE in the 2018–2019 season [3,38,39,40]. It must be pointed out, however, that the design of all the published studies was different from ours; indeed, they were case–control test-negative studies in primary care and hospital settings. Notably, many studies have reported low or no IVE with regard to A(H3N2). The observed antigenic and genetic mismatch between circulating A(H3N2) and the 2018–2019 vaccine strain may well explain this lack of protection against circulating A(H3N2).

As reported in the results of our study, among A(H3N2) viruses, 3C.3a strains were identified mainly in the second part of the season and showed a lower nucleotide identity (range: 96.5%–96.7%) with the vaccine virus than the 3C.2a1b strains. These results are in line with those of several British and American studies [38,39]. Kissling et al. conducted a test-negative study in nine European countries (Croatia, France, Germany, Ireland, the Netherlands, Portugal, Romania, Spain and Sweden) in order to assess VE against influenza A(H3N2) by age and genetic subgroups in the 2018–2019 season. They confirmed that the influenza A(H3N2) clades 3C.2a and 3C.3a co-circulated in Europe and that VE among all ages was −1% (95% CI: −24 to 18) and −26% (95% CI: −66 to 4) respectively. With regard to the subclades: among 15–64 year-olds, IVE against clades 3C.2a1b and 3C.3a was 15% (95% CI: −34 to 50) and −74% (95% CI: −259 to 16), respectively. IVE against clade 3C.3a was very low among 15–64 year-olds, indicating that clade 3C.3a viruses played a major role in the observed IVE against any influenza A(H3N2) in this age group. Increased illness due to antigenically drifted A(H3N2) clade 3C.3a influenza viruses prompted concerns about IVE and vaccine strain selection. Indeed, in the USA, Flannery et al. conducted a test-negative study and used US virologic surveillance and US Influenza Vaccine Effectiveness Network data to evaluate the consequences of this clade. In the 2018–2019 influenza season, A(H3N2) clade 3C.3a viruses caused a high proportion of influenza cases, IVE being 5% (−10% to 19%) against A(H3N2) clade 3C.3a viruses. The predominance of A(H3N2) clade 3C.3a viruses during the latter part of the 2018–2019 season was associated with decreased IVE, supporting need to update the A(H3N2) vaccine component of 2019–2020 influenza vaccines in the northern hemisphere. During the February 2019 consultation, the WHO recommended updating the A(H1N1) component from A/Michigan/45/2015 to A/Brisbane/02/2018 and maintaining B vaccine reference viruses for the 2019–2020 northern hemisphere influenza vaccine [41]. For the first time since 2005 [42] selection of the A(H3N2) component was delayed in order to obtain additional data on changes in the distribution of A(H3N2) viruses, with increased activity due to A(H3N2) clade 3C.3a in some regions, and enabled complete characterization of new clade 3C.3a candidate vaccine viruses. On 21 March 2019, the WHO recommended a clade 3C.3a virus as the A(H3N2) component of the 2019–2020 northern hemisphere vaccines [43,44].

Our study also has a strong point: the fact that it increased vaccination coverage in our cohort of HCWs. This result was observed on comparing the levels of vaccination coverage in the 2018–2019 season with those recorded in the previous season (2017–2018). Moreover, the coverage rate observed in our cohort was distinctly higher than that recorded among HCWs in Italy [7,8,13,14,35]. This finding confirms the utility of the vaccination promotion program implemented in the present study. It therefore appears important to continue this type of promotion through also collaboration with the hospital and local healthcare agencies [11,12]. Indeed, interventions to promote vaccination should also take into account the working demands of HCWs and facilitate their access to vaccination through the creation of several vaccination points within each agency [7,37].

Although the study displayed critical points, it enabled us to identify weak spots and to determine which actions to undertake in order to improve the planning and management of future cohort studies, with a view to meeting the objectives set. Specifically: more time needs to be devoted to the enrolment phase; awareness-raising campaigns need to be implemented in order to improve compliance with the study protocol and to promote the active participation of the subjects enrolled; and there needs to be greater commitment on the part of the research group to carry out surveillance of ILI cases during follow-up (weekly reminders via social media, the implementation of a mobile app and, also phone call).

## 5. Conclusions

Influenza imposes a very heavy burden and causes high direct and indirect costs, some of which are difficult to assess and are probably underestimated. Even if IVE is low, vaccination can avoid many cases of disease, thereby determining a positive impact on national healthcare services and society. Indeed, it is well recognized that, compared with non-vaccination, annual influenza vaccination is cost-effective or cost-saving in several settings and population groups [45,46,47]. In particular, studies conducted on HCWs have shown that seasonal influenza vaccination reduces absenteeism from work during the epidemic period and, consequently, determines a reduction in the direct and indirect costs of influenza [13,48]. In this context, tailor-made pro-vaccination awareness-raising campaigns aimed at increasing vaccine coverage are to be regarded as priority interventions.

Study difficulties and the circulation of drifted variants of A(H3N2) could explain the observed IVE. Therefore, acquiring accurate and updated effectiveness estimates over several influenza seasons could help to better define the many variables that affect vaccine effectiveness and to evaluate the real effectiveness of the awareness-raising measures adopted to increase the vaccination coverage.

## Figures and Tables

**Figure 1 vaccines-08-00119-f001:**
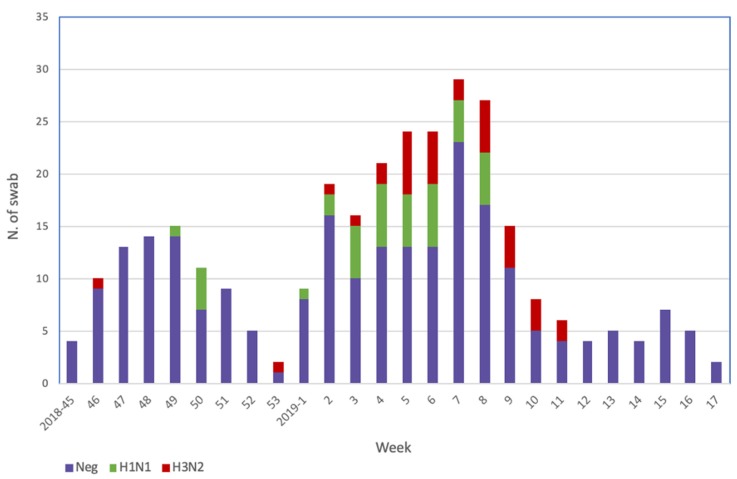
Weekly distribution of samples collected and influenza virus types/subtypes detected.

**Table 1 vaccines-08-00119-t001:** Characteristics and vaccination status of subjects enrolled during the 2018–2019 influenza season.

Age Group (Years)	18–24 (%)	25–34 (%)	35–44 (%)	45–54 (%)	55–64 (%)	≥65 (%)	Total (%)
**Gender**							
F	663 (21.0)	747 (23.6)	412 (13)	767 (24.3)	491 (15.5)	81 (2.6)	3161 (100)
M	250 (18.9)	414 (31.3)	166 (12.6)	211 (16.0)	233 (17.6)	48 (3.6)	1322 (100)
Total	913 (20.4)	1161 (25.9)	578 (12.9)	978 (21.8)	724 (16.1)	129 (2.9)	4483 (100)
**Any chronic condition**							
0	827 (90.6)	1046 (90.1)	510 (88.2)	766 (78.3)	525 (72.5)	90 (69.8)	3764 (84.0)
≥1	86 (9.4)	115 (9.9)	68 (11.8)	212 (21.7)	199 (27.5)	39 (30.2)	719 (16.0)
**Risk factor**							
Smoking	186 (20.4)	254 (21.9)	122 (21.1)	213 (21.8)	178 (24.6)	15 (11.6)	968 (21.6)
**Number of hospitalizations in the previous 12 months**							
0	905 (99.1)	1150 (99.1)	575 (99.5)	957 (97.9)	702 (97)	122 (94.6)	4411 (98.4)
≥1	8 (0.9)	11 (0.9)	3 (0.5)	21 (2.1)	22 (3)	7 (5.4)	72 (1.6)
**Influenza vaccination status in previous season (2017–2018)**							
Vaccinated	67 (7.3)	184 (15.8)	89 (15.4)	120 (12.3)	132 (18.2)	52 (40.3)	644 (14.4)
Unvaccinated	779 (85.3)	889 (76.6)	433 (74.9)	742 (75.9)	483 (66.7)	62 (48.1)	3388 (75.6)
Unknown	67 (7.3)	88 (7.6)	56 (9.7)	116 (11.8)	109 (15.1)	15 (11.6)	451 (10.0)
**Influenza vaccination status in study season (2018–2019)**							
Vaccinated	236 (25.8)	429 (37.0)	175 (30.3)	273 (27.9)	277 (38.3)	69 (53.5)	1459 (32.5)
Unvaccinated	677 (74.2)	732 (63.0)	403 (69.7)	705 (72.1)	447 (61.7)	60 (46.5)	3024 (67.5)
**Vaccine brand among the vaccinated in season 2018–2019**							
Agrippal®	5 (2.1)	-	1 (0.6)	-	2 (0.7)	-	8 (0.5
Fluad®	-	-	-	1 (0.4)	5 (1.8)	21 (30.4)	27 (1.9)
Fluarix Tetra®	122 (51.7)	228 (53.1)	81 (46.3)	157 (57.7)	161 (57.9)	19 (27.5)	768 (52.6)
Vaxigrip Tetra®	109 (46.2)	201 (46.9)	93 (53.1)	114 (41.9)	110 (39.6)	27 (39.1)	654 (44.8)
Unknown	-	-	-	-	-	2 (2.9)	2 (0.1)

**Table 2 vaccines-08-00119-t002:** Characteristics of vaccinated and unvaccinated subjects in 2018–2019 season.

	Vaccinated (%)	Unvaccinated (%)	Total (%)
Total	1459 (32.6)	3024 (67.5)	4483 (100)
**Gender**			
F	946 (29.9)	2215 (70.1)	3161 (100)
M	513 (38.8)	809 (61.2)	1322 (100)
**Any chronic condition**			
0	1193 (31.7)	2571 (68.3)	3764 (100)
≥1	266 (37.0)	453 (63.0)	719 (100)
**Risk factor**			
Smoking	245 (25.3)	723 (74.7)	968 (100)
**Influenza vaccination status in previous season (2017–2018)**			
Yes	583 (40.0)	61 (2.0)	644 (14.4)
No	507 (34.7)	2881 (95.3)	3388 (75.6)
Unknown	369 (25.3)	82 (2.7)	451 (10.1)

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
