# Peer review of "Influenza Vaccination in Italian Healthcare Workers (2018–2019 Season): Strengths and Weaknesses. Results of a Cohort Study in Two Large Italian Hospitals"

_vaccines, 2020, doi:10.3390/vaccines8010119_

Round 1
Reviewer 1 Report
An interesting read and well presented paper.
Of course, a cohort study with a longer follow-up must be considered for a paper in the future. A.o. due to the drifting variants of influenza viruses.
I am looking forward to see such a paper submitted in the future.
Reviewer 2 Report
This manuscript by Donatella Panatto et al. presents an interesting work studying the 2018-19 seasonal influenza vaccine effectiveness in a cohort study performed with healthcare works (HCWs) in two large Italian hospitals.
The study received 308 reports of influenza like illness (ILI), 157 from vaccinated HCWs and 151 from unvaccinated HCWs in the enrolled cohort in season 2018-19. Furthermore, 35 and 37 laboratory-confirmed influenza virus infection were found from the 157 and 151 reported ILI in vaccinated and unvaccinated HCWs respectively. Indeed the vaccine effectiveness of H1N1 and H3N2 viruses discovered in this work showed similar trend in comparison to other studies in 2018-19 season, although it would be fairly difficult to draw any conclusive lines from such a limited number of laboratory-confirmed influenza infection, ILI and participants.
The authors did a fantastic job enrolling 4483 HCWs from two large hospitals, in which there are in total 5700 HCWs working. The pro-vaccine-awareness-raising campaign did also bring the vaccine coverage figure in the HCWs enrolled to 32.5%, a considerably higher value in comparison to other HCW studies or studies on general public.
With the inherited difficulties, limitation and bias of cohort studies appreciated, authors are invited to revise or comment on following issues:
It is, on one hand, impressive to have 4483 out of 5700 HCWs in the two hospitals enrolled in current study. However, on the other hand, it is not surprising to see overall a low compliance with follow-up surveillance especially in the HCWs not vaccinated, given the well observed vaccine scepticism in HCWs all over the world. To some extent, the high percentage of the HCWs enrolled is one of the contributors of the suboptimal ILI report rate. That being said, the initial power and sample size calculation of the study should have been adjusted to compensate this foreseeable risk of low compliance with follow-up surveillance.
As mentioned by authors in discussion section, the weekly reminders, by email or text message, were not enough to guarantee compliance with follow-up surveillance. More direct interaction would have done better, such as weekly follow-up phone calls or questionnaires.
In line 257-260, and line 262-270, authors described the results of laboratory test for influenza virus infection in all the ILI reported in the study. It will be beneficial to the influenza community especially epidemic surveillance network, to have the clinically isolated virus nucleotide sequences accessible through database such as NCBI influenza database or GISAID.
Minor issues:
In line 205-6: “The overall median age was 37 years (26-52)”, what is the numbers in brackets? According to Table 1, they are not the range of ages of all participants. It would be easier to understand if authors give clear explanation here.
In table 1, the row “total” in Gender, Any chronic condition, and Risk factor, etc are exactly the same. It will help the readability of the manuscript by removing this redundancy. The same repetition occurred in Table 2, Table S3 and S4.
Reviewer 3 Report
Review: Influenza vaccination in Italian healthcare workers (2018-2019 season) Strengths and weaknesses. Results of a cohort study in two large Italian hospitals.
Panatto, et al. describe outcomes of vaccination programs in two large Italian hospitals (Milan and Genoa). In the manuscript they provide appropriate background on ILI and optimal vaccination rates among HCWs. As part of the DRIVE program, they seek to describe the infection outcomes and protection associated with influenza vaccination during week 40 of 2018-week 17 of 2019. A strength of the study is utilization of a large cohort of both vaccinated and unvaccinated HCWs during the influenza season. While the cohort is an interesting one, the study design and subsequent description of the data are lacking.
Major comments.
There is no description of the individual occupation of each person considered a HCW. Are these simply people who work in the hospitals, those who see patients directly, administrators without patient contact? This is of great importance because the distinctions of individuals with or without patient contact and later analysis of ILI and vaccine effectiveness may be confounded by these factors. For example, the HCWs who were vaccinated, but had direct patient contact may reasonably be expected to have a higher rate of ILI than similarly vaccinated HCWs who never came in contact with already ill patients. The aggregate of all HCWs together seems to be a disservice to this study and an interpretation that may be made about the actual effective rate of vaccine performance.
The authors describe which vaccines were used in Table 1. The split of Fluarix and Vaxigrip implies that each hospital described in the study may have used a separate formulation. If true, it would be improper to cluster all vaccinated people together. Similarly, additional information regarding the perceived protective effect of each vaccine formulation may be especially useful to the companies that generated them – for example, is the low vaccine efficacy attributable to only one of the most used formulations, implying that there was poor antigenic match despite a similar genetic profile.
Lines 257-270 appear to have no experimental data to support the claims made. Please either upload genome sequences with comparison to a specific reference to support these claims or remove this section. The methods section on genotyping does not imply the ability to detect any type of mutation since the probe sets used are by RT-qPCR incapable of doing so.
Minor comments:
Regarding the Information Technology system (lines 101-102), there is no description of what this includes.
Round 2
Reviewer 2 Report
The manuscript has been improved after revision. However, authors are invited to write the material and methods, and discussion section more concisely.
Author Response
Dear Sirs,
Thank you for considering our manuscript for publication in Vaccines. We found the reviewers’ comments helpful and constructive and feel that the manuscript has been greatly improved and enriched. All reviewers’ comments have been addressed.
Comment: The manuscript has been improved after revision. However, authors are invited to write the material and methods, and discussion section more concisely.
Reply: As required, the discussion section has been revised. Some changes have also been made in the methods section and the conclusion. In the first revision, a more detailed description of the laboratory methods was performed, as required by the reviewer 3. The changes are highlighted using the "Track Changes" function in the revised version of manuscript.
Warm regards,
Donatella Panatto
Reviewer 3 Report
Authors,
Thank you for your extensive and insightful revisions. You have addressed all of my concerns in the appropriate manner. I look forward to future studies that include the additional study design characteristics described in the comments from the other reviewers.
The only limitation of the current version is that the nucleotide sequences have not been released on Genbank. Please ensure these sequences have been released prior to formal publication of this manuscript.
Author Response
Dear Sirs,
Thank you for considering our manuscript for publication in Vaccines. We found the reviewers’ comments helpful and constructive and feel that the manuscript has been greatly improved and enriched. All reviewers’ comments have been addressed.
Point-by-point reply
Reviewer 3
Comment: The only limitation of the current version is that the nucleotide sequences have not been released on Genbank. Please ensure these sequences have been released prior to formal publication of this manuscript.
Reply: Sequences have been submitted to Genbank and will be released once the paper will be published.
Warm regards,
Donatella Panatto